# Overexpression of *SgDREB2C* from *Stylosanthes guianensis* Leads to Increased Drought Tolerance in Transgenic *Arabidopsis*

**DOI:** 10.3390/ijms23073520

**Published:** 2022-03-24

**Authors:** Yun Han, Leilei Xiang, Zhigang Song, Shaoyun Lu

**Affiliations:** Guangdong Engineering Research Center for Grassland Science, State Key Laboratory for Conservation and Utilization of Subtropical Agro-Bioresources, College of Life Sciences, South China Agricultural University, Guangzhou 510642, China; hanyun@stu.scau.edu.cn (Y.H.); 20201002007@stu.scau.edu.cn (L.X.); szg@stu.scau.edu.cn (Z.S.)

**Keywords:** *Stylosanthes guianensis*, *SgDREB2C*, drought, antioxidant enzyme

## Abstract

*Stylosanthes guianensis* is an excellent forage legume in subtropical and tropical regions with drought tolerance, but little is known about its drought tolerance mechanism. Dehydration responsive element binding proteins (DREBs) are responsive to abiotic stresses. A *SgDREB2C* was cloned from *S. guianensis*, while SgDREB2C protein was localized at nucleus. *SgDREB2C* transcript was induced by dehydration treatment. Transgenic *Arabidopsis* overexpressing *SgDREB2C* showed enhanced osmotic and drought tolerance with higher levels of relative germination rate, seedlings survival rate and *F*_v_/*F*_m_ and lower levels of ion leakage compared with WT after osmotic and drought stress treatments. In addition, higher levels of superoxide dismutase (SOD) and ascorbate peroxidase (APX) activities and stress responsive gene (*COR15A*, *COR47*) transcripts were observed in transgenic *Arabidopsis* than in WT under drought stress. These results suggest that SgDREB2C regulated drought tolerance, which was associated with increased SOD and APX activities and stress-responsive gene expression under drought stress.

## 1. Introduction

Drought is one of the major abiotic stresses reducing crop productivity [1]. Thousands of genes are altered in expression in response to drought, leading to morphological, physiological and biochemical changes in plants [1,2]. Drought-responsive gene expression is regulated by dehydration response element binding (DREB) transcription factors that bind to the *cis*-dehydration response element in the promoter region of downstream genes. DREB members are classified into six subgroups, named A-1, A-2, A-3, A-4, A-5 and A-6 in *Arabidopsis* [3]. Among them, DREB1s/CBFs are major proteins involved in the regulation of cold acclimation [4,5,6,7], while DREB2s are involved in plant responses to drought, salinity and heat stress [8,9]. Expression of *DREB2A* and *DREB2B* from *Arabidopsis* plants is induced by dehydration and salt stress [10]. In addition, *AtDREB2A* and *AtDREB2B* have functional redundancy in response to heat stress [11]. Transgenic *Arabidopsis* plants overexpressing *AtDREB2A* show increased drought tolerance and heat tolerance [12], but only slight freezing tolerance [13]. Overexpression of *OsDREB2A* from rice results in enhanced drought and salt tolerance [14,15], while *OsDREB2B* expression leads to improved drought tolerance in transgenic rice [16]. Overexpression of *ZmDREB2A* from *Zea mays* improves heat and drought tolerance in transgenic *Arabidopsis* [8]. *PcDREB2A*-overexpressing *Arabidopsis* plants show increased tolerance to drought and osmotic stress [17]. Overexpression of *EsDREB2B* from *Eremosparton songoricum* leads to increased tolerance to osmotic stress, salt, cold and heat in transgenic tobacco [18].

DREB2C confers tolerance to multiple abiotic stresses in plants. Transcription of *AtDREB2C* is induced by oxidative stress [19], heat [20] and salt stress [21]. Constitutive expression of *AtDREB2C* results in improved thermotolerance and freezing tolerance in transgenic *Arabidopsis* plants [20,21]. It specifically activates HEAT SHOCK FACTOR A3 (HsfA3) and PHYTOCYSTATIN 4 (CYPS4) to regulate heat tolerance in *Arabidopsis* [22,23]. Overexpressing *DREB2C* enhances tolerance to oxidative stress in transgenic *Arabidopsis* by regulating HsfA3, which promotes *APX2* expression [19]. In addition, DREB2C plays an important role in seed germination [24,25]. Transgenic *Arabidopsis* plants overexpressing *AtDREB2C* perform delayed germination and increase abscisic acid (ABA) content [25]. *OsDREB2C* expression in rice is induced by drought and salinity [26]. *MsDREB2C* from *Malus sieversii* is induced by salicylic acid (SA), jasmonic acid (JA), ABA, drought, salt, cold and heat [27]. Its overexpression leads to improved tolerance to drought, heat and cold in transgenic plants [28]. Compared to DREB2A and DREB2B, the role of DREB2C in regulation of drought tolerance was less investigated.

The antioxidant defense system protects plants against drought-induced oxidative damage [29]. It consists of antioxidant enzymes, such as superoxide dismutase (SOD), catalase (CAT), ascorbate peroxidase (APX), and non-enzymatic antioxidants such as ascorbic acid and reduced glutathione [30]. Higher levels of antioxidant enzyme activities are commonly observed in the drought-tolerant species compared with drought-sensitive ones [31,32]. Among four wheat genotypes, drought-resistant genotypes have higher SOD activities and transcript levels of *Mn-SOD* than those in drought-susceptible ones [32]. APX activity is increased in three different drought-tolerant alfalfa varieties, but the highest expression of *MsAPX* is shown in the drought-tolerant variety [29].

*Stylosanthes guianensis* (Aublet) Sw. is a tropical forage legume with drought tolerance that is commonly cultivated in tropical and subtropical areas, but it is sensitive to chilling [33,34]. ABA treatment leads to enhanced chilling tolerance as a result of increased antioxidant enzyme activities [35]. Higher antioxidant enzyme activities are maintained in the chilling-tolerant mutants than in the wild type during chilling stress [36]. *9-cis-EPOXYCAROTENOID DIOXYGENASE* gene (*SgNCED1*) expression is induced in response to drought, salt and low temperature, which is associated with ABA accumulation under abiotic stress in *S. guianensis* [37]. Its overexpression results in promoted ABA synthesis and elevated drought tolerance in transgenic tobacco and stylo plants [33,34]. However, the investigations of drought tolerance in *S. guianensis* plants are limited. An *SgDREB2C* showing greatly induced expression by drought was observed in our unpublished transcriptome analysis, implying that it may play an important role in regulation of drought tolerance in *S. guianensis*. The objectives of this study were to identify the role of *SgDREB2C* in drought tolerance and provide a candidate gene for improvement of drought tolerance in forage legumes. Transgenic *A**rabidopsis* overexpressing *SgDREB2C* was obtained and used for evaluation of drought tolerance. In addition, antioxidant enzyme activities and several stress-responsible genes’ expression in response to drought were detected.

## 2. Results

### 2.1. Characterization of SgDREB2C

A *DREB2C* gene from *S. guianensis* (*SgDREB2C*) was cloned. It contains an open reading frame (ORF) of 1227 bp (GeneBank accession number OK663529) and encodes a deduced polypeptide of 408 amino acids with MW of 44.8 kDa and an isoelectric point at 4.94. Sequence alignment analysis showed that SgDREB2C was most homologous (85.7%) to AipDREB2C (XP_016167062.1) from peanut (*Arachis ipaensis*) in amino acids (Figure 1a). An AP2 DNA binding domain consisting of 64 amino acids (from 82 to 145 aa) was observed in SgDREB2C protein. Phylogenic tree analysis showed that SgDREB2C protein was similar to AtDREB2C (AT2G40340.1) among DREBs in *Arabidopsis* (Figure 1b). Subcellular localization analysis revealed fluorescence of GFP and mCherry in the whole cell, while the fluorescence of SgDREB2C-GFP and NLS-mCherry was observed only in nucleus (Figure 2), indicating that SgDREB2C is localized in nucleus.

### 2.2. Analysis of Spatial Expression of SgDREB2C and Response to Dehydration

*SgDREB2C* transcript was detected in roots, stems, leaves and flowers of *S. guianensis*. The data showed that the highest transcript level was in roots compared with the other tissues (Figure 3a). After dehydration treatment, *SgDREB2C* transcript was induced and reached the maximum level at 2 h, with 16-fold higher expression than in the unstressed control (Figure 3b).

### 2.3. Analysis of Drought Tolerance in Transgenic Plants Overexpressing SgDREB2C

Transgenic *Arabidopsis* plants overexpressing *SgDREB2C* were generated. Seven homozygous lines were selected based on resistance to basta and harvested for further investigations. Three transgenic lines (D3, D5 and D8) with higher levels of *SgDREB2C* transcript were selected for evaluation of drought tolerance (Figure 4). The results showed that the seed germination rate was higher in transgenic plants than that in the wild type (WT) under control conditions. It was decreased significantly in WT plants but unaltered in transgenic plants on the MS medium containing 200 mM mannitol (Figure 5a), which led to higher relative germination rates in transgenic plants as compared with the WT plants (Figure 5b).

The drought tolerance of transgenic lines was evaluated based on survival rate, *F*_v_/*F*_m_ and ion leakage after withholding irrigation. Most WT plants died with a survival rate of 7.4%, while transgenic lines had a survival rate of 70% to 74% after drought stress (Figure 6a,b). *F*_v_/*F*_m_ was decreased and ion leakage was increased greatly after 20 d of withholding irrigation, while higher levels of *F*_v_/*F*_m_ and lower levels of ion leakage were observed in transgenic plants than in WT plants (Figure 6c–e). The results indicated that transgenic plants had increased drought tolerance.

### 2.4. Analysis of Antioxidant Defense System and Stress-Responsive Genes

The antioxidant defense system protects plants against oxidative damage induced by drought. Antioxidant enzyme activities were determined. SOD activity showed no significant difference between transgenic plants and WT plants under control conditions, but higher activities were observed in transgenic plants than in WT plants under drought stress (Figure 7a). CAT activity showed no significant difference between transgenic lines and WT under either control or drought stress conditions (Figure 7b). There was no difference in APX activity between transgenic plants and WT plants under control conditions (Figure 7c). APX activity was greatly reduced in WT after drought treatment, while it was unaltered in transgenic lines after drought treatment (Figure 7c).

Transcript levels of stress-responsive genes were analyzed. *COR15A* and *RD29A* transcript levels showed no difference between transgenic plants and WT under control conditions, but they were greatly induced after drought treatment (Figure 8a,b). Higher levels of *COR15A* transcript were observed in transgenic plants under drought stress (Figure 8a), and transgenic lines D3 and D8 but not D5 had higher *RD29A* transcript levels than WT plants (Figure 8b). The *COR47* transcript level was higher in transgenic lines D3 and D8 than in WT under control conditions. It was greatly induced in transgenic plants after drought treatment, but not altered in WT plants (Figure 8c).

## 3. Discussion

*SgDREB2C* was cloned from *S. guianensis* with an AP2 DNA domain. SgDREB2C protein is located in nucleus, which is consistent with AtDREB2C [21] and MsDREB2C [27]. DREB2s are important transcriptional regulators in response to drought and salt stress [3,9,16]. In our study, the highest transcript level of *SgDREB2C* was observed in roots of *S. guianensis*, and *SgDREB2C* transcript was induced by dehydration. The results implied its potential role in the regulation of drought tolerance. The highest level of *AtDREB2C* transcript was observed in flowers [38], whereas that of *MsDREB2C* was observed in mature leaves [27]. The difference may be associated with gene function in plant species.

Transgenic plants overexpressing *SgDREB2C* showed enhanced osmotic stress and drought tolerance with higher germination rates, survival rates and *F*_v_/*F*_m_ and lower levels of ion leakage compared with WT under drought conditions. The results suggested that SgDREB2C confers drought tolerance. Consistent with these findings, *MsDREB2C* was also significantly induced by drought, and its expression led to increased drought tolerance in transgenic *Arabidopsis* [27]. Photosynthesis is one of the major processes that is significantly affected by drought [39]. *F*_v_/*F*_m_ reflects the photochemical activity of photosystem II (PSII). Drought causes the production of reactive oxygen species (ROS) that can damage PSII and further lead to decreased photosynthesis capability under drought stress [40]. In chickpea, the tolerant genotype has higher *F*_v_/*F*_m_ than the sensitive one under drought stress [41]. In the present study, transgenic *Arabidopsis* plants maintained a survival rate that was associated with higher photosynthesis capability compared with that in WT under drought conditions.

The antioxidant system protects plants against oxidative damage by scavenging ROS under abiotic stress [42,43]. Higher activities of antioxidant enzymes are associated with drought tolerance in diverse plant species. *PcDREB2A*-overexpressing lines have lower ROS levels in transgenic plants under drought stress [17]. Higher activities of SOD and APX were maintained in transgenic *Arabidopsis*, and higher CAT activity was maintained in two lines of *SgDREB2C*-overexpressing transgenic plants compared with WT under drought. Nevertheless, the higher antioxidant enzyme activities are associated with increased drought tolerance in transgenic plants overexpressing *SgDREB2C*.

Stress-responsive genes are regulated by DREBs through binding to the *cis*-acting dehydration-responsive element/C-repeat (DRE/CRT) element in their promoter regions [3,21,28]. *RD29A*, *COR47* and *COR15A* genes are responsive to drought in *Arabidopsis* [44]. The increased drought tolerance in *VvNAC08*-OE [45] and *VyP5CR*-OE transgenic lines [46] and *ataf1-1*, *ataf1-2* mutants of *A. thaliana* [47] is partly associated with higher transcription levels of drought stress-responsive genes *RD29A*, *COR47* and *COR15A*. Higher transcript levels of *COR15A* and *COR47* were maintained in three *SgDREB2C*-overexpressing lines and higher levels of *RD29A* transcript were maintained in two lines compared with WT plants under drought stress. Similarly, the *COR15A* transcript was higher in transgenic *Arabidopsis* overexpressing *AtDREB2C* compared with its level in WT plants, although *COR47* and *RD29A* levels were not affected [21]. Nevertheless, our results indicated that the higher transcript levels of drought-responsive gene expression are associated with the increased drought tolerance in *SgDREB2C* transgenic lines.

In summary, the role of *SgDREB2C* in the regulation of drought tolerance was identified. The *SgDREB2C* transcript was induced by drought, and its constitutive expression led to enhanced drought tolerance in transgenic plants. The increased drought tolerance was associated with the higher levels of stress-responsive gene transcripts and antioxidant enzyme activities compared with the wild type under drought stress.

## 4. Materials and Methods

### 4.1. Plant Growth and Dehydration Treatment

The germinated seeds of *S. guianensis* cv CIAT 184 were sown in 11 cm diameter plastic pots with a mixture of peat and perlite (3:1, *v*/*v*). The seedlings were grown in a greenhouse under natural light with temperatures from 25 to 30 °C for 3 weeks as previously described [36]. Roots, stems, and leaves were harvested for examination of tissue-specific expression, while the flowers from mature plants were harvested. For dehydration treatment, the second leaves from the tops of the seedlings were detached and placed in a laminar flow hood for 4 h for gradual dehydration, followed by immersion in liquid nitrogen for isolation of total RNA.

### 4.2. Cloning and Sequence Analysis of SgDREB2C

Total RNA was isolated from 0.1 g of *S. guianensis* leaves using TRI-Gene RNA Extraction Kit (GenStar Biosolutions Co., Ltd., Beijing, China). Two ug of total RNA was used for reverse transcription by using Evo M-MLV RT Kit with gDNA Clean (Accurate Biology, Changsha, China) in the presence of oligo (dT)_18_. The cDNA was used as a template to amplify the coding sequence of *SgDREB2C* using two primers (forward: 5′-ATGGGTGCTTATGATCAAGGTTCTA-3′ and reverse: 5′-CTCTAATCATCCTTCAACGTTTGCATTATTTCCCTTTTGGAT-3′), and 2×TSINGKE^®^ Master Mix (Tsingke Biotechnology Co., Ltd., Beijing, China). The conserved domain of SgDREB2C protein was analyzed using SMART (http://smart.embl-heidelberg.de/), which was accessed on 20 April 2018. SgDREB2C and its homologous DREB2C sequences from other plants obtained from National Center for Biotechnology Information (NCBI) were aligned using DNAMAN software version 6.0. The phylogenic tree was generated using MEGA version 7.0.26 with the best model JTT+F+I+G4.

### 4.3. Subcellular Localization Assay

The coding sequence of *SgDREB2C* without termination codon was cloned into pYL322-eGFP-N and fused with eGFP. The recombinant vector (35S-SgDREB2C-GFP) and control vector of nucleic localization sequence (35S-NLS-mCherry) were transiently co-transformed into protoplast isolated from 2-week-old rice sheath using PEG-mediated transformation as previously described [48]. The fluorescence was observed by a laser scanning confocal microscope (Leica DM6B, Leica Microsystem CMS GmbH. Mannheim Germany).

### 4.4. Generation of Transgenic Arabidopsis

The coding sequence of *SgDREB2C* was cloned into pCAMBIA3301 driven by CaMV35S promoter. The recombinant vector was introduced into *Agrobacterium tumefaciens* strain EHA105, which was used to transform *Arabidopsis* by the floral dip method [49]. The sterilized seeds of transgenic *Arabidopsis* were placed on MS medium containing 50 mg/L basta to obtain positive transformants. Homozygous transgenic lines (T_3_) were identified.

### 4.5. Evaluation of Drought Tolerance

Fifty sterilized seeds of transgenic lines and wild type were placed on 1/2 MS medium containing 200 mM mannitol or without mannitol as control as previously described [50]. After 4 days, the germinated seeds were counted for calculation of germination rate. Each treatment contained three plates as replicates.

Seven-day-old seedlings grown on 1/2 MS medium were transferred to a 50-hole tray to grow for 7 d in a growth chamber at 22 °C under light of 200 μmol m^−2^ s^−1^. After the seedlings were fully irrigated, plants were withheld from irrigation so that the soil gradually became dry. When the wild type plants showed serious wilting at 21 d after withholding irrigation, ion leakage and maximal photochemical efficiency (*F*_v_/*F*_m_) of photosystem II were measured as previously described [51]. The plants were re-irrigated to allow 3 d of recovery, and surviving plants were counted for calculation of survival rate.

### 4.6. Measurements of Antioxidant Enzyme Activities

Fresh leaves (0.3 g) were ground in 2 mL of 50 mM ice-cold phosphate buffer (pH 7.8) containing 2% (*w*/*v*) PVP and 2 mM EDTA, followed by centrifugation at 14,000 rpm for 30 min at 4 °C. The supernatants were collected for measurements of superoxide dismutase (SOD) and catalase (CAT) activities as previously described [52]. For extraction of ascorbate peroxidase (APX), fresh leaves (0.3 g) were ground in 2 mL of 50 mM phosphate buffer (pH 7.0) containing 1 mM ascorbic acid and 1 mM ETDA, followed by centrifugation as above. APX activity was determined as previously described [52]. One unit of SOD activity was defined as the amount of enzyme required to lead to 50% inhibition of photochemical reduction of ρ-nitro blue tetrazolium chloride (NBT). One unit of CAT and APX activity was defined as the amount of enzyme required for catalyzing the conversion of H_2_O_2_ (extinction coefficient 0.0394 mM/cm) or ascorbic acid (AsA) (extinction coefficient 2.8 mM/cm) within 1 min [52].

### 4.7. Real-Time Quantitative PCR (qPCR)

Total RNA was extracted using TRI Gene reagent (GenStar Biosolutions Co., Ltd., Beijing, China), followed by synthesis of the first strand cDNA using Evo M-MLV Mix Kit (Accurate Biology, Changsha, Hunan, China). PCR was performed using 2 × SYBR^®^ Green Premix Pro Taq HS Premix (Accurate Biology, Changsha, Hunan, China) on the Bio-Rad CFX Manager version 1.6 (Bio-Rad, Hercules, CA, USA). *ACTIN* was used as reference gene to normalized gene expression. qPCR primers are listed in Appendix A. Each sample contained three biological replicates.

### 4.8. Statistic Analysis

All data for these experiments were processed using Excel version 2016 and SPSS software Version 19.0 (IBM, Chicago, IL, USA). The data were tested by one-way analysis of variance (ANOVA) based on least significant difference (LSD) (*p* < 0.05).

## Figures and Tables

**Figure 1 ijms-23-03520-f001:**
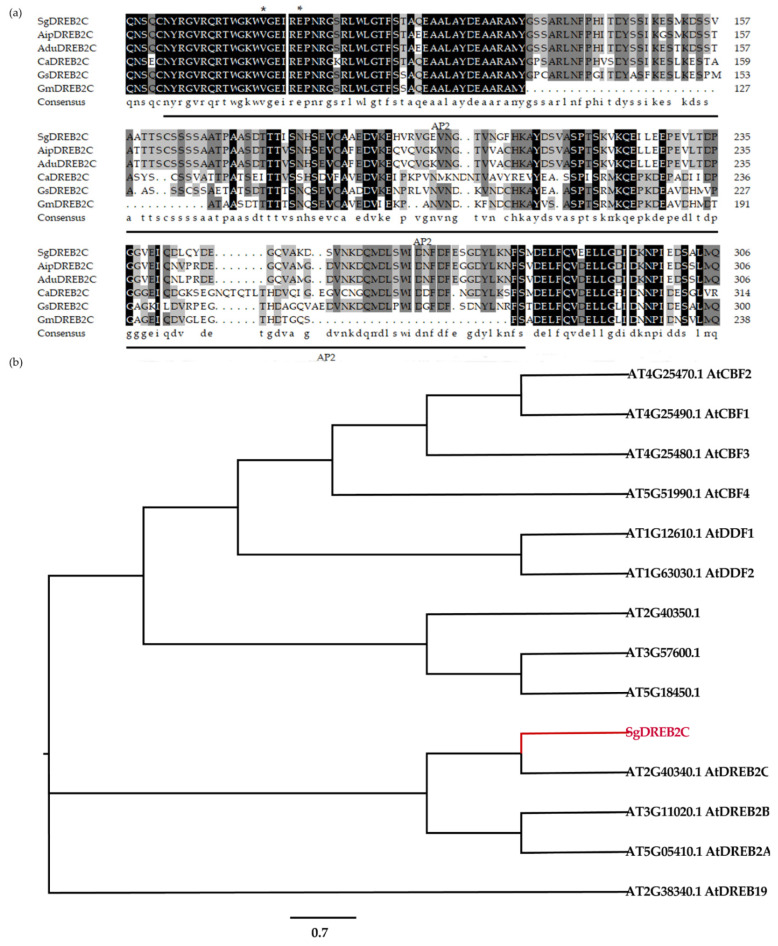
Alignment of multiple sequences and phylogenetic relationships of SgDREB2C and DREBs from *A. thaliana*. (**a**) Multiple sequence alignment of SgDREB2C and DREBs from other plant species including AipDREB2C (XP_016167062.1) from *Arachis ipaensis*, AduDREB2C (XP_015973808.1) from *Arachis duranensis*, AhyDREB2C (XP_025609101.1) from *Arachis hypogea*, CaDREB2C (XP_004491005.1) from *Cicer arietinum*, GsDREB2C (XP_028199704.1) from *Glycine soja*, GmDREB2C (NP_001240005.1) from *Glycine max* and MtDREB2A (XP_003616701.1) from *Medicago truncatula*. The asterisks (*) represent the conserved amino acid. (**b**) Phylogenetic tree analysis of SgDREB2C with other DREBs from different subgroups in *A. thaliana*.

**Figure 2 ijms-23-03520-f002:**
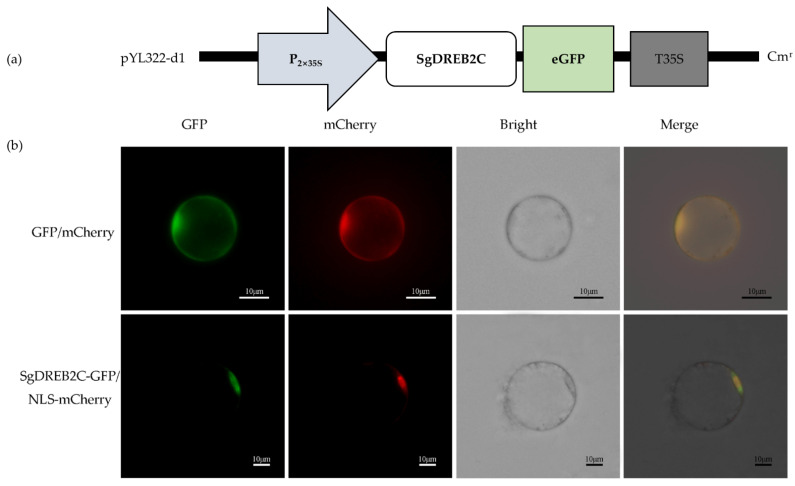
Subcellular localization of SgDREB2C. (**a**) Schematic draft of p35GFP-SgDREB2C construct used for subcellular location assay; (**b**) Confocal images of rice protoplasts co-expressing SgDREB2C-GFP and NLS-mCherry or GFP and mCherry as control.

**Figure 3 ijms-23-03520-f003:**
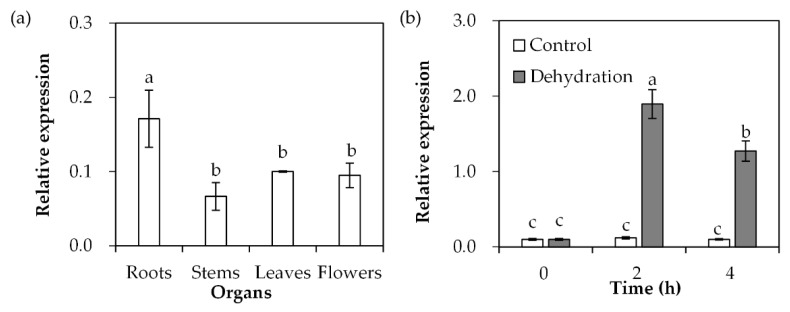
Analysis of *SgDREB2C* transcript in different organs and in response to dehydration treatment. Roots, stems and leaves were sampled from 3-week-old *S. guianensis* seedlings (**a**). The leaves from 3-week-old seedlings were placed in a hood for dehydration treatment to isolate total RNA, which was used to determine the relative expression of *SgDREB2C* using qPCR (**b**), while *ACTIN* was used as reference gene. Means of three replicates and standard errors are presented; the same letter above the column indicates no significant difference at *p* < 0.05.

**Figure 4 ijms-23-03520-f004:**
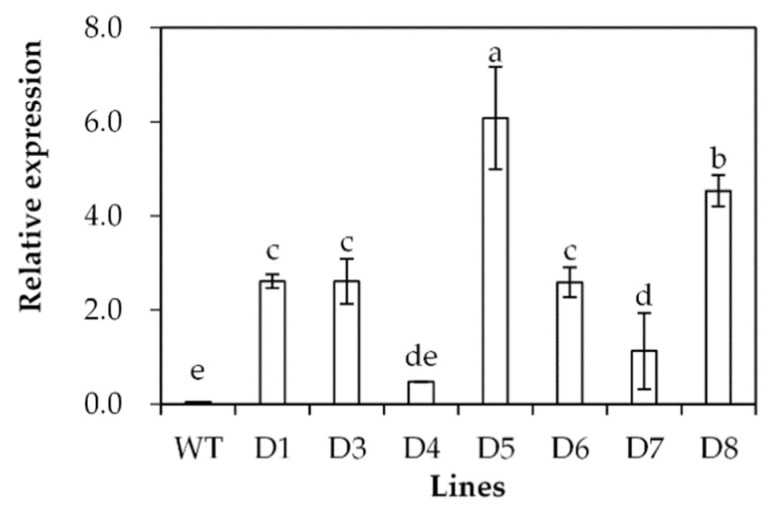
Analysis of *SgDREB2C* transcript in transgenic plants in comparison with the wild type (WT). The relative expression of *SgDREB2C* was evaluated using qPCR, and *ACTIN* was used as reference gene. Means of three replicates and standard errors are presented; the same letter above the column indicates no significant difference at *p* < 0.05.

**Figure 5 ijms-23-03520-f005:**
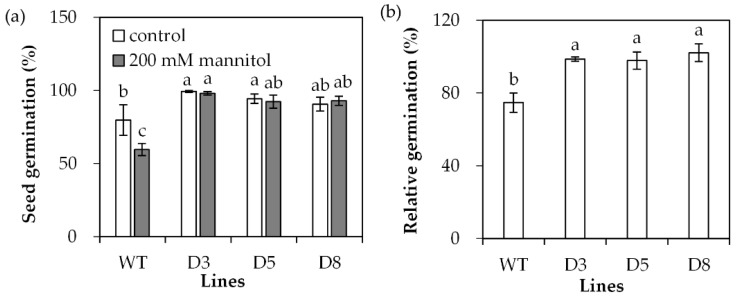
Seed germination in transgenic plants in comparison with the wild type (WT) in response to osmotic stress. Sterilized seeds were placed on 1/2 MS medium containing 200 mM mannitol or without it as control for 4 days for determination of germination rate (**a**), and the data were used to calculate relative germination rate (**b**). Means of three replicates and standard errors are presented; the same letter above the column indicates no significant difference at *p* < 0.05.

**Figure 6 ijms-23-03520-f006:**
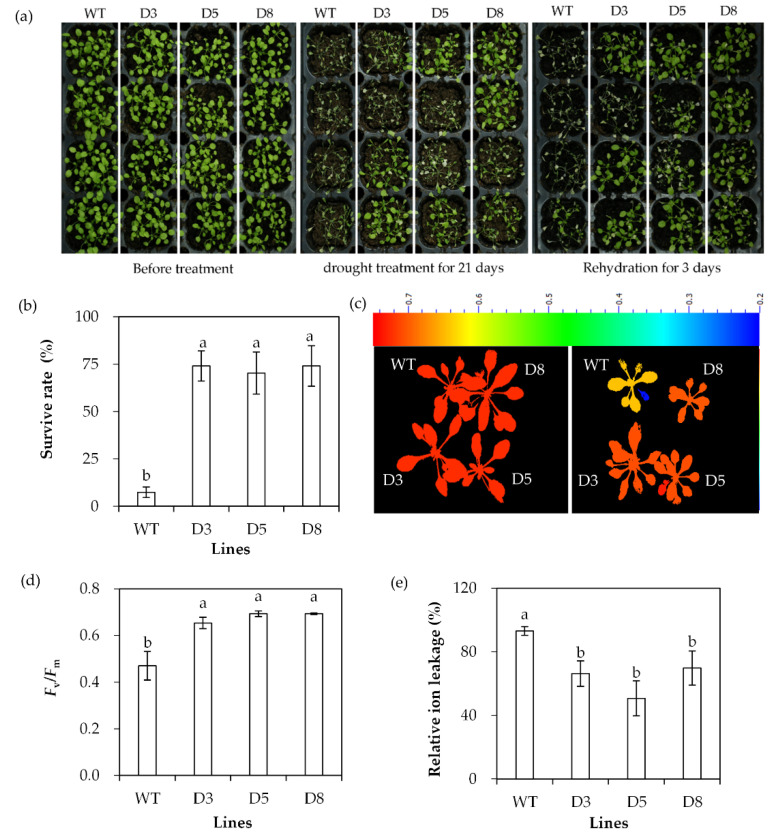
Analysis of drought tolerance in transgenic plants in comparison with the wild type (WT). Two-week-old seedlings were withheld from irrigation for drought treatment. Photos were taken at the time points as indicated in the figure (**a**). Survival rate was determined after 21 days of drought, followed by 3 days of rewatering (**b**). *F*_v_/*F*_m_ (**c**,**d**) and relative ion leakage (**e**) were determined after 14 days of drought. Means of three replicates and standard errors are presented; the same letter above the column indicates no significant difference at *p* < 0.05.

**Figure 7 ijms-23-03520-f007:**
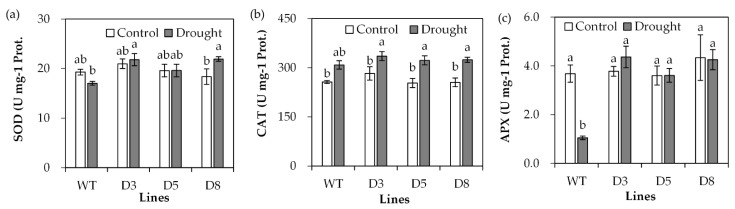
Analysis of antioxidant enzyme activities in transgenic plants in comparison with the wild type (WT). Three-week old seedlings were withheld from irrigation for 14 days (drought) or continuously irrigated as control, and leaves were sampled for measurements of SOD (**a**), CAT (**b**) and APX (**c**) activities. Means of three replicates and standard errors are presented; the same letter above the column indicates no significant difference at *p* < 0.05.

**Figure 8 ijms-23-03520-f008:**
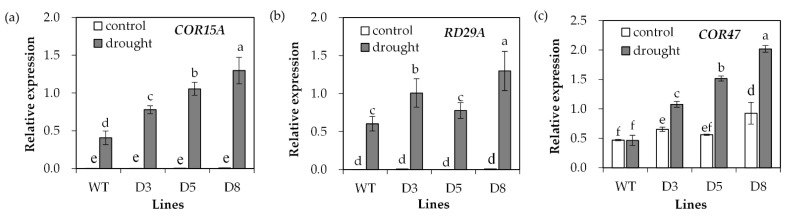
Analysis of drought-responsive marker gene expression in response to drought. Two-week-old seedlings were withheld from irrigation for 14 days (drought) or continuously irrigated as control, and leaves were sampled for determination of the relative expression of drought-responsive genes including *COR15A* (**a**), *RD29A* (**b**) and *COR47* (**c**). Means of three replicates and standard errors are presented; the same letter above the column indicates no significant difference at *p* < 0.05.

## Data Availability

Not applicable.

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
