# Peer review of "Overexpression of SgDREB2C from Stylosanthes guianensis Leads to Increased Drought Tolerance in Transgenic Arabidopsis"

_ijms, 2022, doi:10.3390/ijms23073520_

Round 1

Reviewer 1 Report

The manuscript is poorly written, and data presentation is weak, a

Abstract 

Abstract is carelessly written authors should incorporate their notable findings and adequately connect with the sentences they choose to correspond.

Introduction

  • The introduction section must have a clear hypothesis and significantly develop the second paragraph of your manuscript. Make it more connecting to the problem statement. 
  • Overall there is the repetition of the information, which could be avoided.

Discussion 

  • This section should include more information and references related to the relevant and related works. 

Figure

  •  Figure legends are difficult to interpret.

Conclusions

  • If possible, restructure and carefully edit the conclusion section and add clear information regarding the most noteworthy findings.

Author Response

Abstract is carelessly written authors should incorporate their notable findings and adequately connect with the sentences they choose to correspond.

Response: Abstract has been revised.

Introduction

  • The introduction section must have a clear hypothesis and significantly develop the second paragraph of your manuscript. Make it more connecting to the problem statement. 
  • Overall there is the repetition of the information, which could be avoided.

Response: A hypothesis has been added in the text, and Introduction has been also revised.

Discussion 

  • This section should include more information and references related to the relevant and related works. 

Response: Discussion has been revised.

  •  Figure legends are difficult to interpret.

Response: The legends have been revised.

  • Conclusions
  • If possible, restructure and carefully edit the conclusion section and add clear information regarding the most noteworthy findings.

Response: The conclusion has been revised.

Response: It has been added in the text.

Reviewer 2 Report

Article review

"Overexpression of SgDREB2C from Stylosanthes guianensis leads to increased drought tolerance" by Yun Han , Leilei Xiang , Zhigang Song and Shaoyun Lu

This work is devoted to the production of Arabidopsis plants transgenic for the gene

 SgDREB2C, which was isolated from the legume Stylosanthes guianensis.

 This plant easily tolerates drought, which was the reason for the use of its genes responsible for resistance to dehydration. As a result, the authors managed to obtain Arabidopsis lines that grow well on mannitol and express the target gene in all parts of the plant. In addition, the authors analyzed the activities of the key antioxidant enzymes SOD, ASA, and catalase in these plants during drought and tried to correlate these results with data on the expression of the COR15A, COR47, and RD29A genes that respond to drought.

In the course of reading this work, I would like to make some comments and clarifications, they will benefit the manuscript and make it more understandable.

* In the title of the work, you must indicate the object of the work, as long as you have subjected only Arabidopsis to transgenesis, indicate this in the title

* In the introduction, everything is good and a lot, but bring to the goal of the work. So you wrote - The objective of this study was to identify the role of SgDREB2C 79 in drought tolerance. The study will provide a candidate gene for improvement of 80 drought tolerance in forage legume.

But in the goal it is necessary to cover everything that you do in the work, this is to improve the perception of the material.

*In materials and methods - for enzymes, put the nomenclature

* On the basis of which 200 mM mannitol was modeled in the work

* check captions Fig. 3 and 4

* in Figure 6(c) this is the photochemical activity of the leaves, this needs to be said more in the discussion

* in the discussion - you associate with drought resistance only with two enzymes, describe the contribution of catalase to the resistance of the obtained lines of transgenic plants, in addition, you analyze the expression of the RD29A gene in transgenes during drought, in the lines D3 and D8 there is a significant expression of this gene, why do you not discuss it and correlate stability with it too

*is it possible to understand that transgenic plants, judging by all the results, are insensitive to stress, for the sake of completeness, it would be good to give data on the generation of ROS in these plants.

Author Response

* In the title of the work, you must indicate the object of the work, as long as you have subjected only Arabidopsis to transgenesis, indicate this in the title

Response: The title has been changed.

* In the introduction, everything is good and a lot, but bring to the goal of the work. So you wrote - The objective of this study was to identify the role of SgDREB2C in drought tolerance. The study will provide a candidate gene for improvement of 80 drought tolerance in forage legume. But in the goal it is necessary to cover everything that you do in the work, this is to improve the perception of the material.

Response: The objectives have been revised, and more information has been added.

*In materials and methods - for enzymes, put the nomenclature

Response: Full name of the enzymes has been added.

* On the basis of which 200 mM mannitol was modeled in the work.

Response: A citation has been added in the text, which is the basis of 200 mM mannitoal being used (L304).

* check captions Fig. 3 and 4

Response: The legends have been revised.

* in Figure 6(c) this is the photochemical activity of the leaves, this needs to be said more in the discussion

Response: It has been discussed in the revision.

* in the discussion - you associate with drought resistance only with two enzymes, describe the contribution of catalase to the resistance of the obtained lines of transgenic plants, in addition, you analyze the expression of the RD29A gene in transgenes during drought, in the lines D3 and D8 there is a significant expression of this gene, why do you not discuss it and correlate stability with it too

Response: Three antioxidant enzyme and RD29A.

*is it possible to understand that transgenic plants, judging by all the results, are insensitive to stress, for the sake of completeness, it would be good to give data on the generation of ROS in these plants.

Response: Thanks for the good suggestion. ROS generation has not been added due to the limit of re-submission deadline. We will measure it in our further studies in the future.

Round 2

Reviewer 1 Report

Authors have significantly improved the manuscript; therefore, the manuscript can be published after a thorough English language test,